# Molecular Regulators of Embryonic Diapause and Cancer Diapause-like State

**DOI:** 10.3390/cells11192929

**Published:** 2022-09-20

**Authors:** Abdiasis M. Hussein, Nanditaa Balachandar, Julie Mathieu, Hannele Ruohola-Baker

**Affiliations:** 1Department of Biochemistry, University of Washington, Seattle, WA 98195, USA; 2Institute for Stem Cell and Regenerative Medicine, University of Washington, Seattle, WA 98109, USA; 3Department of Biotechnology, School of Bioengineering, SRM Institute of Science and Technology, Chennai 603203, India; 4Department of Comparative Medicine, University of Washington, Seattle, WA 98109, USA

**Keywords:** embryonic diapause, quiescence, cancer stem cells, metabolism, epigenetics, mTOR

## Abstract

Embryonic diapause is an enigmatic state of dormancy that interrupts the normally tight connection between developmental stages and time. This reproductive strategy and state of suspended development occurs in mice, bears, roe deer, and over 130 other mammals and favors the survival of newborns. Diapause arrests the embryo at the blastocyst stage, delaying the post-implantation development of the embryo. This months-long quiescence is reversible, in contrast to senescence that occurs in aging stem cells. Recent studies have revealed critical regulators of diapause. These findings are important since defects in the diapause state can cause a lack of regeneration and control of normal growth. Controlling this state may also have therapeutic applications since recent findings suggest that radiation and chemotherapy may lead some cancer cells to a protective diapause-like, reversible state. Interestingly, recent studies have shown the metabolic regulation of epigenetic modifications and the role of microRNAs in embryonic diapause. In this review, we discuss the molecular mechanism of diapause induction.

## 1. Introduction

Diapause is a reversible state of suspended embryonic development that is associated with delayed blastocyst implantation. Embryos from over 130 mammalian species can enter this dormant state, in which the blastocysts survive unimplanted in the uterus for a prolonged period of time [1]. Two types of diapause, obligate and facultative, have been described in vertebrates (reviewed by [2]). Obligate diapause occurs in each gestation of a species and serves to ensure that offspring are born in favorable environmental conditions. On the other hand, facultative diapause is an optional halt in development at the implantation stage in response to circumstances/environment. Facultative diapause can be caused, for example, by metabolic stress due to lactation. In mice, the blastocyst remains in a diapause state as the newly born pups nurse because the suckling stimulus promotes an increased secretion of prolactin, which downregulates ovarian estrogen release [3,4]. Elimination of the stimulus induced by suckling through the removal of pups results in an increase in ovarian estrogen and initiates embryonic reactivation and implantation of the embryo. Facultative diapause can, therefore, be used as a reproductive strategy to avoid metabolic stress due to nursing overlapping litters [5,6]. Diapause can last from a few days to several months, depending on the species [7]. In mice, embryonic diapause can last up to 36 days in vivo [8]. Recent work characterizing the diapause observed in wild roe deer embryos (diapause coincides with the hunting season) has shed important light on the molecular regulation of diapause in vivo in the wild [9].

Fertilization takes place when a sperm cell fuses with an egg cell, resulting in the formation of the single diploid cell zygote. The zygote, which contains chromosomes from both the sperm and the ovum, undergoes mitotic division with no significant growth. The process of cell division leads to the formation of early and late blastocysts and, finally, the implantation of the embryo into the uterine wall (Figure 1). However, in embryonic diapause, the embryo is arrested at the blastocyst stage prior to implantation (Figure 1). In many vertebrates, unfavorable environmental conditions can lead to a complete block of embryonic development at the blastocyst stage; however, development resumes when the stress is removed [3,10,11]. Importantly, recent findings have suggested that cancer stem cells utilize a diapause-like quiescent stage to avoid stress/insult (chemotherapy)-induced apoptosis [12,13].

## 2. Diapause Metabolism

Embryonic diapause is associated with reduced metabolic activities, including protein and DNA synthesis and carbohydrate metabolism [1,14,15,16]. We and others have shown that pluripotent stem cells grown in vitro in states corresponding to pre-implantation (naïve) and post-implantation (primed) stages exhibit different metabolic and epigenetic (discussed in 3.3) profiles [17,18,19,20,21]. Naïve pluripotent stem cells are able to perform oxidative phosphorylation, fatty acid oxidation and glycolysis, while primed cells rely exclusively on glycolysis as a source of energy [17,22]. A metabolic switch is thus necessary during the developmental progression from the pre- to the post-implantation embryo. Transcriptomic and metabolic analyses have previously shown that some of the metabolic pathways enriched in the stage between the pre- and post-implantation stages, the diapause stage, include glycolysis, lipolysis, fatty acid oxidation, pyruvate, and cholesterol metabolism [23,24]. These and previous studies have also suggested that autophagy is activated in a dormant blastocyst rather than a reactivated blastocyst [25]. Furthermore, autophagy is also activated in dormant cancer cells [12,13]. The activation of autophagy might provide essential nutrients, allowing the prolonged survival of embryos during diapause. However, since diapause is associated with decreased overall metabolism, it is not clear why some metabolic pathways are enriched in diapause embryos. Below, we describe some possibilities for this quandary.

Lipolysis, the breakdown of triacylglycerol (the lipid storage components) into free fatty acids and glycerol, is increased in diapause compared to pre-implantation blastocysts [24]. Lipolysis has been connected with adaptation to starvation [26], and breakdown by lipolysis can lead to the accumulation of free fatty acids and phosphatidylcholines (PC). PCs have been proposed to play a critical role in the activation of the NF-κB pathway [27]. These data suggest that lipolysis, which is upregulated in diapause, may aid in cell survival through the NF-κB pathway [24,28,29,30]. Furthermore, the NF-κB pathway regulates autophagy by activating the expression of genes that are critical for the formation of the autophagosome, including membrane lipid biosynthesis, which is required for autophagosome vesicle production [31,32,33,34]. Since autophagy leads to the breakdown of lipids, the NF-κB pathway may act as a rheostat (as seen in Drosophila; [35]) coordinating the function of lipid metabolism and autophagy in the diapause stage. Furthermore, glycerol released from triacylglycerol breakdown can enter the glycolytic pathway as DHAP (dihydroxyacetone-3-phosphate), in line with observations of increased glycolytic pathway activity in diapause [24].

A recent paper may explain the importance of free fatty acids (generated by lipolysis) in diapause. Under conditions of metabolic stress, fatty acid oxidation (FAO) was found to be critical for embryonic stem cell survival. This process is controlled by the transcription factor Tfcp2L1/Lbp9, which is involved in the maintenance of the naïve pluripotent state [23]. Tfcp2L1 promotes FAO by activating Cpt1a, an enzyme that aids in the transfer of free fatty acids to the mitochondria [17]. Upon the deletion of Tfcp2L1, the authors found an increase in glycolysis and glutamine catabolism. The Tfcp2L1–Cpt1a axis promotes blastocyst survival in a diapause-like condition induced by the inhibition of mTOR. When mTOR was inhibited, more cell death was observed in the Tfcp2L1 KO cells than in the wild-type cells, suggesting that Tfcp2L1 is required in cell survival upon mTOR inhibition [23]. This concurs with the previous findings that post-implantation, primed pluripotent stem cells (PCS) with low Tfcp2L1/Lbp9 levels cannot survive mTOR inhibition [17,21] and suggests Tfcp2L1/Lbp9 as a master regulator between diapause and activated post-implantation blastocyst stages; diapause PSC can survive mTOR inhibition due to Tfcp2L1/Lbp9-Cpt1a axis, while primed PSC with reduced transcription of Lbp9 cannot [21]. Interestingly Tfcp2L1/Lbp9 was previously found to be one of the PRC2 target genes for the earliest critical repressive histone H3K27me3 marks in embryonic stem cells. Based on the recent studies, the H3K27me3 based repression of Lbp9 gene is expected to take place during re-activation of diapause embryo. The findings suggest the key embryonic transcription factor Tfcp2L1/Lbp9 maybe the culprit that answers long lasting question in early embryonic development; it maybe the missing master regulator for the mitochondrial metabolic switch observed in naïve-to-primed ESC transition [17,18,22]. Further studies on the timing of Lbp9 repression by PRC2, and its role in diapause-like state in cancer will be informative. 

Furthermore, recent studies may explain the source of lipolysis in diapause. Diapause blastocysts have been shown to use lipid droplets to survive while they remain in a dormant state [8]. The authors suggested that lipid droplets are not essential at the pre-implantation stage or during implantation but only in the diapause state. Lipid droplets have been previously shown to be critical in early development [36,37] and knocking out one of the lipid droplets associated proteins, Plin2, led to an exit from pluripotency [38]. In Plin2 KO, enhanced lipid hydrolysis leads to the disruption of mitochondrial cristae, resulting in decreased acetyl-CoA. This decrease leads to the acetylation of H3K27 and hence causes an exit from the pluripotent state. Since lipid droplets are required for early development, the authors investigated the role of lipid droplets in the naïve-to-primed transition. Wild-type cells transitioned from the naïve to the primed state, whereas the Plin2 KO cells showed differentiation. These results reveal that Plin2-negative cells lead to an exit from pluripotency, indicating a role of lipid droplets in maintaining the pluripotent state. These data suggest that lipid droplets play a critical role in maintaining embryonic diapause and, plausibly, are the substrates for the identified increase in lipolysis during diapause.

## 3. Molecular Mechanisms of Diapause Regulation

The regulation of embryonic diapause is under maternal control [1]. A multitude of factors has been implicated in the control of embryonic diapause, including transcription factors, epigenetic regulation, uterine microRNAs, and the mTOR pathway. Here, we highlight the role of the maternal or embryonic regulators that were shown to play a key role in embryonic diapause and connect their activity with new findings that might inform us of their possible mechanisms of action.

### 3.1. The Transcription Factor Hesx1

The highly conserved homeobox transcriptional repressor Hesx1 (Homeobox gene expressed in stem cells 1) was recently found to play a key role in early development and in embryonic diapause [39]. However, previous work showed that *Hesx1*^−/−-^ mouse embryos die perinatally [40,41], suggesting that a lack of Hesx1 might not be detrimental for in utero development. Hesx1 is expressed in the mouse pre-implantation embryo, and previous research suggested that Hesx1 might be required for human embryonic stem cells (ESCs) to remain pluripotent [42]. It was proposed that Hesx1 might repress differentiation-related genes to maintain the pluripotent state [43]. This discrepancy could be explained by Hesx1 being required for stress-induced quiescence and diapause. If the pregnancy is stress-free, Hesx1 mutants proceed normally through uterine development; however, Hesx1 mutants may fail to normally proceed through the protective diapause stage during stress. To support this, a recent paper reported that Hesx1 was essential to re-initiate normal mouse development after the diapause stage in vivo [39].

### 3.2. Micro-RNA LET-7

Several other factors are known to regulate embryonic diapause in rodents in cases of unfavorable environmental conditions, such as nutritional deprivation or energetic stress [7,44]. MicroRNAs have been implicated in embryonic diapause [7,45,46]. MicroRNAs are highly conserved, small (~22 bp nucleotide), non-coding RNAs that play critical roles in the post-transcriptional regulation of gene expression. They regulate their target genes primarily by binding the 3′ UTR region of the protein-coding mRNA, which results in translational repression or transcript destabilization [47]. MicroRNAs are crucial for animal development [48], as they play critical roles in diverse cellular processes, such as proliferation, differentiation, cell cycle progression, and apoptosis [47,49,50].

MicroRNAs were shown to be differentially expressed in pre-implantation naïve and post-implantation primed human and mouse pluripotent stem cells [17,51]. Both diapause, which is an intermediate step between the naïve and primed stages, and reactivated dormant blastocysts are associated with changes in microRNA expression [45]. The let-7 family is more highly upregulated in mouse diapause embryos than in reactivated embryos [45]. A member of this family, microRNA let-7a was found to block the implantation of embryos, keeping embryos in a dormant state [45]. Targets of the microRNA let-7a include genes involved in the mid-gestation embryonic program that are essential for proliferation and growth [52]. However, it was previously unclear how let-7a was able to block the implantation of embryos and which targets were involved in this process.

A recent paper provided potential clarification of let-7 function. Maternal origin let-7a vesicles reversibly induced embryonic diapause, which was associated, as expected, with decreased proliferation and reduced DNA synthesis in the embryo [53]. The overexpression of pre-let-7a blastocysts cultured ex vivo remained viable longer—for more than 12 days—compared with control blastocysts, which had shrunken in size or degenerated by day 7 [53]. By comparing the gene expression profiles of in vivo dormant blastocysts, in vitro let-7a-induced dormant blastocysts, and activated blastocysts, the authors showed similar profiles for the in vivo and let-7a-induced dormant blastocysts, which were distinct from those of activated blastocysts [53]. The transcript expression of the transcription factor c-Myc and Akt [54,55] were shown to be more reduced in the let-7-induced dormant blastocysts than in untreated control blastocysts [53]. Immunofluorescence staining showed a decreased expression of *Rictor*, a component of mTORC2, in diapause embryos compared with that in estrogen-induced reactivated embryos [53]. Using Targetscan, the authors identified *Rictor* as a potential target gene for let-7a, consistent with their immunofluorescent data. Both mTOR inhibition and Myc depletion were previously found to induce a diapause-like state in mice [24,56,57]. Mechanistically, let-7a was found to inhibit c-Myc, mTORC1, and mTORC2. Pharmacological manipulation decreased the phosphorylation of mTORC1 but not mTORC2 targets when c-Myc was inhibited, whereas the inhibition of mTORC2 had no effect on c-Myc. These data suggest that c-Myc is upstream of mTORC1 but not mTORC2.

Although mTORC1 inhibition is important for the induction of paused pluripotency [58], the inhibition of both mTOR complexes 1 and 2 is needed for the complete induction of diapause embryos [56]. The inhibition of both complexes with INK-128 or RapaLink-1 was found to significantly extend blastocyst survival *ex vivo*, whereas mTORC1 inhibitors, such as rapamycin, slightly extended blastocyst survival ex vivo [56]. These data suggest that both mTOR complexes are required for the successful induction of embryonic diapause. Furthermore, previous work showed that TORC2 inhibition induces lipolysis (see above), which is critical for activating NF-κB activity as a rheostat in the system [24]. The role of let-7 in inhibiting Rictor in the diapause state offers an explanation for how the mTOCR2 pathway is downregulated in diapause (Table 1).

Other microRNAs are potentially involved in regulating embryonic diapause. We analyzed microRNAs that were either highly upregulated or downregulated and the status of their target genes in diapause embryos compared to those of reactivated embryos [51]. Using microarray data of diapause and reactivated embryos [45] and RNA-seq of diapause and post-implantation embryos [24], 379 consistent connections were found between 38 microRNAs and 274 of their target genes [51]. Genes were identified that were differentially expressed in diapause in several studies [24,45,53] (Table 1), and the microRNAs that target them were identified on the basis of predictions from Targetscan [59].

The transcription factor and stress response gene EGR1 is essential for early reproductive development [60]. *Egr1* expression is highly upregulated in embryonic diapause, and the microRNAs that target *Egr1* (microRNA 199 and microRNA 181) are suppressed in diapause [24,51]. The expression of EGR1 is increased in this quiescent state because it might play a role in ensuring that diapause embryos remain viable for implantation, as knockdown of *Egr1* results in a reduction in implantation sites [61]. The diapause-enriched glutamine transporters *Slc38a1* and *Slc38a2* show the same trends in that the microRNAs that target them (miRNA-181 and miRNA-199, respectively) for degradation are downregulated in diapause compared with pre-implantation embryos [24,51]. Knocking out *Slc38a1* and *Slc38a2* resulted in the loss of viable embryos [24]. Using chemical inhibitors against these glutamine transporters in diapause embryos caused an exit from the diapause state [24].

### 3.3. Epigenetic Remodeling in Diapause

During embryonic development, an extensive chromatin remodeling of promoters and enhancers accompanies the transition from the pre-implantation to the post-implantation stage [62,63,64,65]. Both DNA and histone methylation in mammals are epigenetic modifications that can impact gene expression and affect cell fate. Naïve pre-implantation ESCs are characterized by global DNA hypomethylation and lack of PRC2-dependent H3K27me3 marks on histones, while primed post-implantation pluripotent cells display high levels of DNA methylation and require PRC2 complex activity [17,66,67,68]. Epigenetic remodeling has also been shown to play an important role in regulating embryonic diapause [24,56,69]. According to proteomic studies, proteins involved in chromatin remodeling are increased upon reactivation of the dormant blastocyst [69]. More recently, immunofluorescence staining in the ICM of diapause blastocysts highlighted a significant reduction in histone marks associated with active transcription—H4K5/8/12 acetylation and H3K36me2 [56], H4K16 acetylation [24,56], and H3K4me3 [70]—compared with those in pre-implantation blastocysts. By contrast, the repressive histone modification mark H3K27me3 was significantly increased in the ICM of diapause blastocysts compared to that in reactivated blastocysts [70]. These data suggest that the chromatin landscape of embryonic diapause is remodeled, and that diapause has a distinct epigenetic signature that encompasses an increase in repressive marks and a significant reduction in activating marks. Chromatin remodeling signature that is similar to in vitro embryonic diapause is also seen in vivo, in animals that experience diapause. Unfavorable environmental conditions, such as starvation, crowding, and heat stress, can induce a diapause-like dauer stage in *Caenorhabditis elegans* [71,72]. In *C. elegans*, animals that have gone through the dauer stage have distinct and dramatically different histone modification marks compared with animals that have not gone through the dauer stage [73]. Both acetylated and methylated histone 3 lysine 4 (H3K4Ac and H3K4me)—marks that are associated with active transcription—were decreased significantly in post-dauer animals compared with those in animals that did not transit through the diapause stage [73].

As mentioned above, during early embryonic development, naïve and primed pluripotent cells exhibit different epigenetic signatures [17,21,68], and pre-implantation development is associated with a global reduction in DNA methylation [19]. DNA methylation marks are reduced as cells progress from fertilization to the formation of the inner cell mass [67], but the DNA methylation levels increase as cells transition to the post-implantation primed pluripotent stage [67]. Accordingly, previous studies have shown that DNA methyltransferase, DNMT1, has an essential function in human and mouse primed pluripotency [74,75]. These findings suggest that DNA methylation is not required in mouse naïve ESCs, but it is essential in both mouse and human primed pluripotency. However, it is not clear whether DNA methylation is required in vivo in embryonic diapause. The de novo DNA methyltransferase DNMT1 was found to be a Hesx1-binding protein (perhaps as expected since Hesx1 is a repressor [76]). Mutations that lead to disruption of Hesx1 DNA binding properties had phenotypes similar to those observed in Hesx1-deficient mice [76]. It is possible that DNA methylation is also essential in embryonic diapause, and this might explain why homozygous mutant Hesx1 embryos are unable to carry out a normal diapause state.

Changes in metabolites and metabolism in diapause embryos can play an important role in controlling their epigenetic state. For example, several mitochondrial metabolites are rate-limiting substrates in epigenetic reactions and could explain the specific epigenetic signatures observed in diapause [18,21]. In particular, it was shown recently that glutamine transporters are essential for the H4K16Ac-negative diapause state [24]. Future studies will allow the depiction of the interplay between metabolism and epigenetics during diapause in more detail.

### 3.4. mTOR and the Maternal Control of the Diapause State

Mechanistic target of rapamycin (mTOR), also known as mammalian target of rapamycin (mTOR), is a serine/threonine protein kinase and a major regulator of cellular growth and metabolism in response to growth factors and nutrient status [77,78]. In mammals, mTOR is composed of two protein kinase complexes—mTOR complex 1 and mTOR complex 2—and both contain unique and common components that are essential for their functions. mTOR has been shown to be essential not only in metabolism but also self-renewal and exit from the pluripotent state both in humans and mice [79,80,81,82]. mTOR is also essential in early development, as knocking out various components of mTOR complexes leads to embryonic lethality because of impaired proliferation [83,84,85,86,87,88]. These findings emphasize that mTOR is indispensable for the growth of early embryos.

Given the key roles that mTOR complexes play in cell growth and proliferation, it is surprising that while mTOR is required for the primed pluripotent stage, it is not required for the naïve ESC stage [80]. It follows that mTOR inhibition would disrupt the transition of pre-implantation embryos to the post-implantation stage [80].

Recent work has revealed that the inhibition of the mTOR pathway or the downregulation of the transcription factor Myc can induce diapause in mice in the pre-implantation blastocyst stage and the diapause-like state in mouse embryonic stem cells [24,56,89]. Although both mTOR inhibition and Myc depletion are required for the induction of diapause, the signals that reduce Myc or influence how mTOR is downregulated in diapause and the signals that reactivate both these signaling factors are poorly understood in all mammals both in vitro and in vivo [1,9,16,24,56,57,70,90,91,92]. However, recent data showed that the starvation-induced LKB1—AMPK axis inhibited the mTOR pathway and thereby activated a diapause-like state in mouse ESCs. This in vitro starvation model has a gene expression signature that mimics in vivo diapause state, suggesting that starvation can activate diapause through LKB1–AMPK on the basis of the repression of the mTOR pathway [24].

While starvation induces embryonic diapause, a recent study on diapause in wild roe deer reiterated that the opposite is true as well: amino acids were needed to activate mTORC1 from low proliferation during diapause to resume embryonic development [9]. The abundance of these amino acids coincided with the exit of the embryo from the diapause state [9]. These amino acids include mTORC1-priming and mTORC1-activating amino acids, such as glutamine. Further studies are needed to shed light on the role of glutamine and glutamine transporters on mTORC1 regulation. It would also be interesting to investigate whether there is a synergistic effect of glutamine and other mTORC1-activating and mTORC1-priming amino acids as embryos are released from the diapause state.

Transient mTOR inhibition (10uM Torin1 for 3 h) has also been shown to convert primed pluripotent stem cells into a naïve pluripotent state [93]. This process of converting primed cells into a naïve state was plausibly accomplished by the nuclear translocation of the mTOR-responsive transcription factor TFE3 [80,93]. Although diapause is an intermediate state between the naïve and primed stages, it is not clear whether inhibiting mTOR in the primed stage forces cells to enter the diapause stage on their way to becoming naïve cells. It would be interesting to investigate the identity of cells that are transiting the primed stage using transcriptomics and compare them with gene expression data of embryonic diapause.

Since inhibition of mTOR is a critical event in the induction of embryonic diapause, understanding the regulation of mTOR is imperative for understanding diapause. While some of the regulation can occur directly in the embryo, it is known that maternal uterine regulation of diapause is essential as well. The abovementioned work with maternally derived let-7, regulating Rictor and thereby mTOR, is an important example of uterine diapause blastocyst control. In addition, other components, such as muscle segment homeobox (MSX) genes *Msx1* and *Msx2*, were found to be uterine factors associated with blastocyst dormancy [94]. Msx1 expression is high in diapause [6], and its expression is reduced upon the reactivation of the dormant blastocyst with the replacement of estrogen or LIF [95]. Mice with a uterine-specific deletion of *Msx1* and *Msx2* are not able to maintain uterine quiescence, which is required for the embryo to survive while in the diapause state [94], and this deletion is associated with increased NF-κB-mediated inflammation [96]. Mechanistically, the role of MSX in the induction of diapause is not entirely clear. Findings from a recent paper suggest that Msx1 inhibits the PI3K–mTOR mediated signaling pathway, as the phosphorylation levels of PI3K, mTOR, and Akt proteins were reduced in the presence of Msx1 or Msx1 with its interactor PIASy in HUVEC cells [97]. Although this is an exciting finding, additional studies are needed to confirm the inhibitory role of transcription factor Msx1 in uterine quiescence and to further understand whether Msx1 from the uterus affects the PI3K–mTOR metabolic pathway both in the uterus and the embryo.

## 4. Embryonic Diapause and Its Relevance in Cancer Stem Cells

Cancer can arise in many different ways, but in general, a subpopulation of many aggressive cancer cells has characteristics of both stem cells and cancer cells. These cancer stem cells (CSCs) in a tumor, like stem cells in a normal environment, have the capacity to self-renew and give rise to progenitor cells that can become committed to different lineages of cancer cells within the tumor [98]. CSCs could arise from normal stem, progenitor, or differentiated cells due to environmental alterations or genetic mutations [99]. Like normal stem cells, CSCs can transit into dormancy, in which they are known to be resistant to most clinical anti-cancer treatments, such as chemotherapy, and can contribute to tumor recurrence [100].

Studies on embryonic diapause from our group and others may have clinical implications in cancer and can potentially contribute to treatments that target cancer stem cells. One of the main factors that can lead to tumor relapse and cause the failure of chemotherapy is the ability of cancer stem cells to enter a quiescent state [101,102,103]. This is because cancer stem cells remain dormant but alive, and they still have the potential to exit the quiescent state at some point, comparable to diapause embryonic stem cells. Hence, CSCs are considered to exist in a diapause-like state. Recent studies have suggested the cytotoxic treatment of tumor cells followed by diapause-inhibiting therapeutics, such as the modulation of Myc activity, can keep CSCs out of the dormant state and, finally, eliminate them [12,13]. This is a viable strategy because it proposes using cytotoxic drugs to target the cancer cells, combined with the activation of the dormant state, allowing CSCs to exit their “hideout” and become sensitized to chemotherapeutic agents. The authors observed that inducible Myc upregulation can signal these dormant/diapause-like CSCs to exit their quiescent stage to resume normal proliferation [12]. When Myc activation was combined with the inhibition of the serine/threonine cyclin-dependent kinase 9 (CDK9), which is the kinase subunit of positive transcription elongation factor b (P-TEFb), the authors showed enhanced chemosensitivity in chemotherapy-persistent tumor cells [12].

Another potential way to terminate dormancy is the chemical inhibition of the glutamine transporter Slc38a1/2 to cause the exit from the diapause state [24]. Depletion of glutamine transporter activity was shown to escape the diapause state in embryogenesis [24]. The depletion of Myc was found to induce a diapause-like state and proliferative arrest [57], and the inhibition of Myc in tumor cells caused resistance to cytotoxic treatments [57]. Similarly, we predict that inhibiting the glutamine transporter Slc38a1/2 in cancer will result in cancer stem cells exiting the quiescent CSC stage. These findings related to Myc and Slc38a1/2 can be used to design CSC-specific drug cocktails in combination with existing cancer treatments to drive CSCs out of dormancy and eradicate tumor recurrence. This can be achieved by inhibiting the glutamine transporter and activating Myc, allowing CSCs to proliferate again, enter a biosynthetically active state, and exit the dormant state. This will allow cytotoxic treatments to target cancer cells and reduce their ability to survive in an environment that hinders their capacity to enter dormancy and evade chemotherapeutic drugs. Similarly, future studies will show if PRC2 based repression of Tfcp2L1/Lbp9 is critical for re-activation of dormant cancer cells causing tumor recurrence (a major cause for mortality in cancer patients). In particular, it will be interesting to test if the potential re-activation can be blocked by eliminating PRC2 activity in Lbp9 promoter by Cas9-based targeting of a designed protein inhibitor EBdCas9 to the precise locus [104].

## 5. Conclusions

On the basis of the findings from recent papers discussed in this review, we now can draw an updated model for the molecular regulation of embryonic diapause (Figure 2). Many of the identified regulators culminate in mTOR and Myc inhibition. Specifically, recent work has revealed four regulators of the mTOR pathway: the starvation-induced Lkb1–AMPK axis, MSX transcription factors, Let-7, and glutamine transporters (Slc38a1/2). In addition, Tfcp2L1/Lbp9 plays a key role in coordinating the essential metabolic switch needed for diapause exit. Today, it is not known if these regulators act independently or in concert.

The activities of several transcription factors (Myc, Msx, and EGR1) change upon the induction of diapause, and the expression levels of the glutamine transporters Slc38a1/2 are upregulated. Increased expression of glutamine transporters is also required for embryos to remain in the diapause stage, as chemical and genetic inhibition of these transporters results in the exit from the diapause stage [24]. Furthermore, the microRNA let-7 was also shown to inhibit the transcription factor Myc. The depletion of Myc causes the embryo to enter diapause and is associated with the downregulation of biosynthetic pathways in diapause. Another transcription factor that is essential for normal embryonic diapause is Hesx1. Mutations in Hesx1 result in the failure of embryos to continue embryonic development after implantation.

Future studies will reveal the details of the molecular mechanism of these components and their interactions. These studies will be impactful both from the basic science and from the cancer biology point of view. It will be interesting in the future to test whether the other diapause regulators, in addition to Myc and mTOR (Slc38a1/2, LKB1/AMPK, Egr1. Tfcp2L1/Lbp9 and Msx) are critical in CSC diapause-like state (Figure 2). If so, the regulation of these pathways may be helpful in future therapies since the elimination of diapause-like cancer stem cells is essential for successful 21st-century cancer treatment.

## Figures and Tables

**Figure 1 cells-11-02929-f001:**
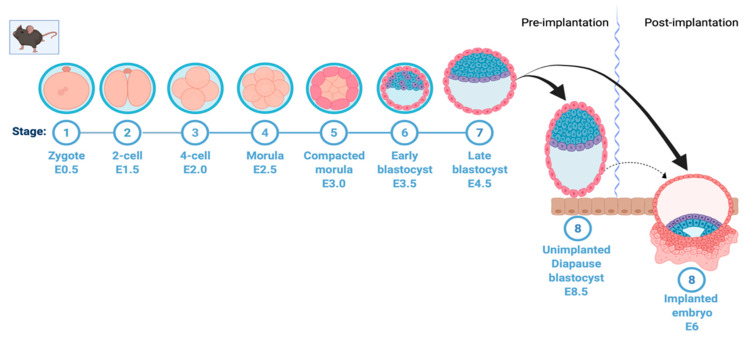
Mouse early embryonic development and induction of diapause. A simplified depiction of mouse early embryonic development showing formation of zygote, two-cell, four-cell, and morula stage, followed by early blastocyst on embryonic day 3.5. In normal development, this is followed by formation of the late blastocyst on embryonic day 4.5 and uterine implantation of the embryo on day 6. This process is halted in embryonic diapause at the blastocyst stage on day 3.5; the embryo hatches from the zona pellucida, stays loosely attached to the uterine wall and remains viable without implanting. This figure was created with Biorender.com (accessed on 25 May 2022).

**Figure 2 cells-11-02929-f002:**
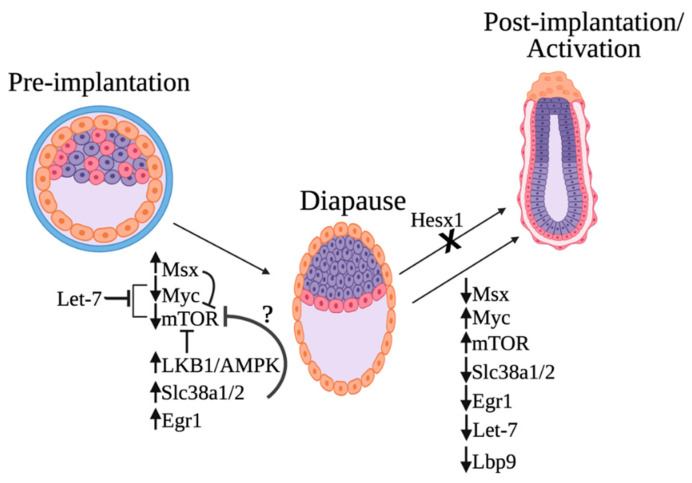
Proposed model of embryonic diapause molecular regulation. Induction of diapause is caused by the elevated levels of Msx1, Egr1, and the glutamine transporters Slc38a1/2 and the downregulation of Myc and mTOR. The starvation-induced Lbk1–AMPK axis is also critical for mTOR inhibition. The microRNA let-7 is highly upregulated in diapause and induces embryonic diapause by inhibiting both Myc/mTORC1 and mTORC2. Msx can potentially repress mTOR, and Slc38a1/2 might also inhibit the activation of the mTOR complex. Downregulation of Tfcp2L1/Lbp9 and metabolic switch takes place during exit from diapause. Furthermore, mutation of the transcriptional repressor Hesx1 in diapause disrupts exit from embryonic diapause and continuation of embryonic development after implantation. This figure was created with biorender.com (accessed on 1 May 2022).

**Table 1 cells-11-02929-t001:** Changes in microRNAs and their target genes in diapause embryos compared with those in reactivated and post-implantation embryos.

MicroRNA	Target Gene	Context++ Score Percentile
mmu-miR-199a-3p	* Slc38a1 *	75
mmu-miR-199b-3p	* Slc38a1 *	75
mmu-miR-199a-3p	* Egr1 *	87
mmu-miR-199b-3p	* Egr1 *	87
mmu-miR-181-5p	* Egr1 *	85
mmu-miR-181-5p	* Slc38a2 *	84
mmu-let-7a-5p	* Rictor *	83
mmu-let-7b-5p	* Rictor *	83
mmu-let-7c-5p	* Rictor *	83
mmu-let-7d-5p	* Rictor *	85
mmu-let-7e-5p	* Rictor *	83
mmu-let-7f-5p	* Rictor *	82
mmu-let-7g-5p	* Rictor *	84
mmu-let-7i-5p	* Rictor *	84
mmu-let-7k	* Rictor *	83

Red: upregulation; blue: downregulation.

## Data Availability

Not Applicable.

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
