# Peer review of "Molecular Regulators of Embryonic Diapause and Cancer Diapause-like State"

_cells, 2022, doi:10.3390/cells11192929_

Round 1

Reviewer 1 Report

In this manuscript, Hussein et al concisely reviewed the recent progress in the regulation of diapause, and its relavance in cancer stem cells. Generally, this topic is of great interest for the fields of embryo development, stem cells, and cancers.

Major concern:

1. The authors should distinguish the induction, maintenance, and exit of diapause, particularly the indcution and maintenance. In Figure 2 and related text, it is necessary to distinguish whether the upregulation of diapause regulators is the consequence or the cause of diapause. Which are the inducer of diapause? And which are required for the maintenance of diapaused embryos?

Minor concerns:

1. The authors used naïve and primed embryonic stem cells (ESCs). While naïve ESC is widely used, primed ESC is an inappropriate term. Rather, primed pluripotency is used to described the pluripotency status in mouse epiblast stem cells and conventional human ESCs.

2. There are many grammar errors and confusing sentences throughout the manuscript. For example, “Naive pre-implantation ESC is characterized by a global genome hypomethylation and lack of H3K27me3 marks in histone, while primed post-implantation ESC display high levels of DNA methylation and require PRC2 compex”, “DNA methylation levels are reduced as cells progress from fertilization to the induction of the inner cell mass in vitro equivalent naïve stage, which is marked by a reversible genome-wide DNA hypomethylation (Theunissen et al., 2016).”, “Afterward, DNA methylation levels increase as cells transition from the pluripotent stem cell naïve to the primed stage (Theunissen et al., 2016).”, and “Mechanistically, let-7a was found to inhibit c-MYC, mTORC1 and mTORC2 and pharmacological manipulation of C-MYC and mTORC2 has shown decreased phosphorylation of only mTORC1 targets when C-MYC is inhibited, whereas inhibition of mTORC2 had no effect on C-MYC, suggesting that C-MYC is upstream of only mTORC1 and not mTORC2.”

Author Response

Reviewer 1:

In this manuscript, Hussein et al concisely reviewed the recent progress in the regulation of diapause, and its relavance in cancer stem cells. Generally, this topic is of great interest for the fields of embryo development, stem cells, and cancers.

Major concern:

  1. The authors should distinguish the induction, maintenance, and exit of diapause, particularly the indcution and maintenance. In Figure 2 and related text, it is necessary to distinguish whether the upregulation of diapause regulators is the consequence or the cause of diapause. Which are the inducer of diapause? And which are required for the maintenance of diapaused embryos?

Response: Thank you for the comments. We have now distinguished the induction and maintenance of the diapause stage. We have also modified figure 2 to clear any confusions. Specifically, we moved all the events/genes that were originally below the diapause stage now below the arrow leading to the diapause. We also clearly described the genes that are the inducers of diapause (see Figure 2 and its related text).

Minor concerns:

  1. The authors used naïve and primed embryonic stem cells (ESCs). While naïve ESC is widely used, primed ESC is an inappropriate term. Rather, primed pluripotency is used to described the pluripotency status in mouse epiblast stem cells and conventional human ESCs.

Response: Thank you for the suggestion. We have now changed the use of primed ESC to primed pluripotency throughout the manuscript.

  1. There are many grammar errors and confusing sentences throughout the manuscript. For example, “Naive pre- implantation ESC is characterized by a global genome hypomethylation and lack of H3K27me3 marks in histone, while primed post-implantation ESC display high levels of DNA methylation and require PRC2 compex”, “DNA methylation levels are reduced as cells progress from fertilization to the induction of the inner cell mass in vitro equivalent naïve stage, which is marked by a reversible genome-wide DNA hypomethylation (Theunissen et al., 2016).”, “Afterward, DNA methylation levels increase as cells transition from the pluripotent stem cell naïve to the primed stage (Theunissen et al., 2016).”, and “Mechanistically, let-7a was found to inhibit c- MYC, mTORC1 and mTORC2 and pharmacological manipulation of C-MYC and mTORC2 has shown decreased phosphorylation of only mTORC1 targets when C-MYC is inhibited, whereas inhibition of mTORC2 had no effect on C- MYC, suggesting that C-MYC is upstream of only mTORC1 and not mTORC2.”

Response: We apologize the mistakes and have now corrected the grammar errors and the confusing sentences.

Reviewer 2 Report

The authors submitted a review for publication, which lists the so-called master regulators of diapause, in their opinion. By definition, master regulators are top proteins in the hierarchy of regulatory signaling pathways with deterministic function (Chan and Kyba, 2013 doi: 10.4172/2157-7633.1000e114). Unfortunately, the authors failed to clearly demonstrate the high-hierarchical functions of all the proteins they mentioned in their work. For this reason, the authors should revise the title of the review, which would more adequately reflect the essence of the work done, or conduct a deeper analysis of the literature, which would allow using the proposed title of the work. Thus, the proposed concept of this mini-review is very poorly developed, and for this reason, the presentation of a new opinion and a fresh look at the conclusion is not obvious.

I suggest that the authors reconsider the structure of the review to make it more conceptual, since, in general, the analysis of the latest data literature has been carried out satisfactorily:

In the chapter “Diapause metabolism”, the presented results can be summarized into a more complete view than the authors presented. As an example, lipid metabolism, autophagy and NF-kB pathway can be brought together. Autophagy is known to be a key signature of dormant embryonic stem cells, dormant stem cells, and dormant cancer cells. Autophagy functions through the formation of autophagosomes, which require membrane lipid biosynthesis. Autophagy, especially selective autophagy (lipophagy, mitophagy), promotes lipid catabolism. Lipid metabolism and autophagy are regulated by NF-kB pathway. The biological meaning of lipid metabolism in the context of autophagy should be shown. And correlate these results with other data described in this chapter.

In the chapter “Molecular mechanism of diapause induction”, the authors write that “the role of Hesx1 in diapause hasn’t been thoroughly investigated. A recent paper found Hesx1 to be essential during diapause.” This means that the role of Hesx1 as a diapause inducer has not yet been determined and may not be one. From here, it is also not entirely clear the role of DNMT1, a HESX1 binding protein, as a diapause inductor, indicated in this chapter. DNMT1 is the methyltransferase, for this reason, discussions about DNMT1 and Hesx1 proteins should be made in the chapter “Epigenetic remodeling in diapause”. In this case, the chapter will be able to introduce a new addition regarding the epigenetic regulation of diapaused embryonic stem cells.

The chapter “mTOR function in diapause” does not reveal any new ideas regarding the mTOR signaling pathway, but reiterates the interpretations available in the literature regarding also myc protein. The indication of the mTOR pathway in this review is useful from the standpoint that mTOR inhibition is a key molecular event in the induction of diapause. And further, the authors should present data regarding the transcription factors MSX1 and MSX2 as maternal controls of dormant blastocysts, which target the mTOR pathway directly in the blastocysts, as external regulatory mechanism of mTOR suppression. And in accordance with this, designate the chapter as the maternal control of the diapause state.

In the chapter “Micro-RNA LET-7”, the authors write that “in vitro let-7a-induced dormant blastocysts, and activated blastocysts, the authors show similar profiles for the in vivo and let-7a-induced dormant blastocysts, distinct from 298 activated blastocysts”. These results mean that Micro-RNA LET-7 may be, by definition, a master regulator of diapause state, and for this reason this chapter should be referred to as “molecular mechanism of diapause induction” or “master regulators”.

The comment to the chapter “Embryonic diapause and its relevance in Cancer Stem Cells” refers to the question why in this chapter the authors did not describe the role of the protein factors, analyzed in the review, in the dormant state of cancer cells. Comparing dormant cancer cells to diapaused stem cells makes sense if there can be a common biological principle between them. Literature data (Rehman et al., 2021) suggest that the transcriptional profile of dormant cancer cells and diapaused embryonic stem cells is similar. In this context, it makes sense to discuss the potential role of proteins, involved in diapause, in dormant cancer cells.

Author Response

Reviewer 2:

The authors submitted a review for publication, which lists the so-called master regulators of diapause, in their opinion. By definition, master regulators are top proteins in the hierarchy of regulatory signaling pathways with deterministic function (Chan and Kyba, 2013 doi: 10.4172/2157-7633.1000e114). Unfortunately, the authors failed to clearly demonstrate the high-hierarchical functions of all the proteins they mentioned in their work. For this reason, the authors should revise the title of the review, which would more adequately reflect the essence of the work done, or conduct a deeper analysis of the literature, which would allow using the proposed title of the work. Thus, the proposed concept of this mini-review is very poorly developed, and for this reason, the presentation of a new opinion and a fresh look at the conclusion is not obvious.

I suggest that the authors reconsider the structure of the review to make it more conceptual, since, in general, the analysis of the latest data literature has been carried out satisfactorily:

  1. In the chapter “Diapause metabolism”, the presented results can be summarized into a more complete view than the authors presented. As an example, lipid metabolism, autophagy and NF-kB pathway can be brought together. Autophagy is known to be a key signature of dormant embryonic stem cells, dormant stem cells, and dormant cancer cells. Autophagy functions through the formation of autophagosomes, which require membrane lipid biosynthesis. Autophagy, especially selective autophagy (lipophagy, mitophagy), promotes lipid catabolism. Lipid metabolism and autophagy are regulated by NF-kB pathway. The biological meaning of lipid metabolism in the context of autophagy should be shown. And correlate these results with other data described in this chapter.

Response: We thank the reviewer for their thoughtful suggestions. We have now made the changes suggested by the reviewer connecting autophagy, lipid metabolism and the NF-kB pathway in diapause.

  1. In the chapter “Molecular mechanism of diapause induction”, the authors write that “the role of Hesx1 in diapause hasn’t been thoroughly investigated. A recent paper found Hesx1 to be essential during diapause.” This means that the role of Hesx1 as a diapause inducer has not yet been determined and may not be one. From here, it is also not entirely clear the role of DNMT1, a HESX1 binding protein, as a diapause inductor, indicated in this chapter. DNMT1 is the methyltransferase, for this reason, discussions about DNMT1 and Hesx1 proteins should be made in the chapter “Epigenetic remodeling in diapause”. In this case, the chapter will be able to introduce a new addition regarding the epigenetic regulation of diapaused embryonic stem cells.

Response: We have now clarified the role of Hesx1 in diapause based on recent papers. Furthermore, the epigenetic remodeling section has now been moved down below the Molecular mechanism of diapause regulation and DNMT1 and Hesx1 proteins are now being discussed in the chapter “Epigenetic remodeling in diapause”.

  1. The chapter “mTOR function in diapause” does not reveal any new ideas regarding the mTOR signaling pathway, but reiterates the interpretations available in the literature regarding also myc protein. The indication of the mTOR pathway in this review is useful from the standpoint that mTOR inhibition is a key molecular event in the induction of diapause. And further, the authors should present data regarding the transcription factors MSX1 and MSX2 as maternal controls of dormant blastocysts, which target the mTOR pathway directly in the blastocysts, as external regulatory mechanism of mTOR suppression. And in accordance with this, designate the chapter as the maternal control of the diapause state.

Response: We have now modified the title of the section from “mTOR function in diapause” to mTOR and the maternal control of the diapause state. We have also moved the section “The transcription factors MSX1 and MSX2” and combined it with this section.

  1. In the chapter “Micro-RNA LET-7”, the authors write that “in vitro let-7a-induced dormant blastocysts, and activated blastocysts, the authors show similar profiles for the in vivo and let-7a-induced dormant blastocysts, distinct from 298 activated blastocysts”. These results mean that Micro-RNA LET-7 may be, by definition, a master regulator of diapause state, and for this reason this chapter should be referred to as “molecular mechanism of diapause induction” or “master regulators”.

Response: We thank the reviewer for this great suggestion. To make it easier for readers, we have now numbered the sections/chapters and moved the “Micro-RNA LET-7” chapter under “the molecular mechanisms of diapause regulation”.

  1. The comment to the chapter “Embryonic diapause and its relevance in Cancer Stem Cells” refers to the question why in this chapter the authors did not describe the role of the protein factors, analyzed in the review, in the dormant state of cancer cells. Comparing dormant cancer cells to diapaused stem cells makes sense if there can be a common biological principle between them. Literature data (Rehman et al., 2021) suggest that the transcriptional profile of dormant cancer cells and diapaused embryonic stem cells is similar. In this context, it makes sense to discuss the potential role of proteins, involved in diapause, in dormant cancer cells.

Response: We thank the reviewer for this suggestion and have modified the section.

Round 2

Reviewer 2 Report

The authors used all my comments proposed to correct the mini review. I thank the authors for their work, which, in my opinion, improved the original structure of the text. But, unfortunately, I have to ask the authors to change the title of the review, because it contains an error. The heading "Molecular regulators of Embryonic and Cancer Diapause" implies that cancer cells undergo diapause or that there is a phenomenon of cancer diapause, but this is not the case. You need to correct the title, or use a phrase "diapause-like".

Author Response

We thank the reviewer for this very good point. Indeed, we should make the title change the reviewer is suggesting. Hence the final title of the review is:

Molecular regulators of embryonic diapause and cancer diapause-like state